# FAST PROXIES FOR LLM ROBUSTNESS EVALUATION

**Tim Beyer, Jan Schuchardt, Leo Schwinn, Stephan Günnemann**
Technical University of Munich & Munich Data Science Institute
`{tim.beyer,j.schuchardt,l.schwinn,s.guennemann}@tum.de`

## ABSTRACT

Evaluating the robustness of LLMs to adversarial attacks is crucial for safe deployment, yet current red-teaming methods are often prohibitively expensive. We compare the ability of fast proxy metrics to predict the real-world robustness of an LLM against a simulated attacker ensemble. This allows us to estimate a model's robustness to computationally expensive attacks without requiring runs of the attacks themselves. Specifically, we consider gradient-descent-based embedding-space attacks, prefilling attacks, and direct prompting. Even though direct prompting in particular does not achieve high attack success rate (ASR), we find that it and embedding-space attacks can predict ASRs well, achieving $r_p = 0.87$ (linear) and $r_s = 0.94$ (Spearman rank) correlations with the full attack ensemble while reducing computational cost by three orders of magnitude.

## 1 INTRODUCTION

As the capabilities of large language models advance, ensuring their robustness and reliability becomes increasingly critical. To this end, frontier models undergo extensive adversarial testing and red-teaming to identify vulnerabilities before deployment (OpenAI, 2023; Dubey et al., 2024).

However, state-of-the-art red-teaming methods are computationally expensive, as finding adversarial prompts is a challenging combinatorial optimization problem over discrete natural language. Here, model-agnostic approaches require prohibitive computational resources (Zou et al., 2023; Chao et al., 2023), whereas more efficient attack algorithms tend to be model-specific and struggle to transfer across architectures (Liao & Sun, 2024). Moreover, reliable red-teaming with strong attacks still demands significant manual effort in tailoring the attack algorithm to a specific model (Andriushchenko et al., 2024; Li et al., 2024). As a result, large-scale red teaming approaches require thousands of GPU hours (Samvelyan et al., 2024), making thorough safety evaluations prohibitively expensive in most research settings.

To address this problem, we propose a scalable alternative: low-cost proxies for real-world threat models. These proxies enable LLM robustness evaluation without needing to run highly expensive automated attack suites against the model. As an example of such an attack suite, we use a "synthetic red-teamer" ensemble comprising six distinct LLM attack methods, which we evaluate on 33 open-source models across 300 harmful prompts. We leverage substantial computational resources and aggregate more than 7M jailbreak attempts. The data suggest that model robustness in adversarial settings can be predicted through inexpensive approaches.

Our main contributions are as follows:

- We investigate whether inexpensive proxies — including direct prompting, prefilling, and embedding space attacks — can predict robustness against strong adversarial red teaming.
- We demonstrate that robustness can be predicted within model families (e.g., different Llama 3 versions) and across model families (e.g., Llama and Mistral).
- Finally, we show that by estimating the most robust model checkpoint during training, proxy attacks can aid adversarial model alignment across different training regimes(e.g., circuit breaking or adversarial training).

## 2 SYNTHETIC RED-TEAMER

To emulate a strong attacker, we create a *synthetic red-teamer* by ensembling six common attack algorithms (listed in Table 1). All attacks are run using the recommended hyperparameters (see also

Appendix B) and simulate a strong red-teamer with significant computational resources ($\approx$ 30 H100-minutes per prompt). We evaluate each algorithm in the *many-trial* setting, where all candidate prompts (including intermediate steps) are tried on the victim model. Thus, for each harmful prompt in the dataset, a model is attacked by 727 different input prompts. If *any* of the prompts succeed, we count the attack as successful. While some algorithms (e.g., AmpleGCG and PAIR) perform many-trial attacks by default, others, such as GCG and BEAST, generally only use the final attack prompt to generate a harmful response. The many-trial setting makes attacks strictly more powerful, at the cost of increased compute.

Table 1: Attacks in synthetic red-teamer ensemble & how many jailbreak candidates they generate per prompt.

| Attack Name | Candidates |
|---|---|
| AmpleGCG (Liao & Sun, 2024) | 200 |
| AutoDAN (Liu et al., 2023) | 100 |
| BEAST (Sadasivan et al., 2024) | 40 |
| GCG (Zou et al., 2023) | 250 |
| HumanJB (Mazeika et al., 2024) | 112 |
| PAIR (Chao et al., 2023) | 25 |
| Total | 727 |

## 3 PROXY METHODS

We aim to find an inexpensive and fast approach that can reliably predict a model's real-world robustness. Finding such a proxy for robustness could dramatically reduce the cost of robustness evaluations, make it easier to compare models across and within families, and efficiently select promising checkpoints during defense training. To this end, we consider three candidate approaches:

**Embedding Space Attacks.** Schwinn et al. (2023; 2024) recently proposed a white box attack that operates in continuous token embedding space, rather than the discrete input vocabulary. This framework—while impractical for real-world attacks, where most threat models assume a black box setting with string-level input—provides an extremely fast way to attack models in a white box setting, and can be used e.g., to adversarially train LLMs (Xhonneux et al., 2024).

**Prefilling.** Prefilling attacks (Vega et al., 2023; Andriushchenko et al., 2024) rely on injecting a prefix to the beginning of the victim model's response to the harmful prompt - typically using an affirmative response prefix. As this level of access is also provided by some private models (e.g., the Claude family (Anthropic, 2024)), it represents a realistic attack vector even for hosted models.

**Direct.** *Direct* prompting is the simplest possible baseline: We simply use an unmodified harmful prompt from the dataset and sample a single greedy generation, which is then judged.

## 4 EXPERIMENTAL EVALUATION

We conduct experiments to determine how well the attack success rates of inexpensive proxy methods (direct ASR, prefilling ASR, embedding-space ASR) predict robustness against real-world red-teaming approaches, which we simulate using our strong synthetic red-teamer from Section 2 across various training and attack scenarios. In addition to directly comparing the different ASRs, we compute Pearson correlation ($r_p$) to quantify linear correlation between proxy ASR and ensemble ASR. We further compute Spearman ($r_s$) and Kendall rank ($\tau$) correlation to understand whether the order of any two models w.r.t. proxy ASR is predictive of their order w.r.t. ensemble ASR. For the full details of our experimental setup, see Appendix B. For additional results see Appendix C.

### 4.1 COMPARING WITHIN-FAMILY MODELS

Popular base models are often fine-tuned for particular use cases, such as chatting (Tunstall et al., 2023), helpfulness (Zhu et al., 2023), or tool use (Teknium et al., 2024). We are interested in comparing the safety of several post-trained model versions. In Figure A, we evaluate different derivatives of Llama 3 8B Instruct. Spearman and Kendall rank correlation coefficients $r_s$ and $\tau$ of direct prompting are greater or equal than those of the other proxy attacks. We observe that direct ASR is close to 0 for multiple models, which impedes a good linear fit ($r_p$ of 0.62) between direct ASR and ensemble ASR. This $r_p$ is smaller than those of prefilling and embedding space attacks. Thus, even for within-family comparisons, the simplest and fastest attack appears like a suitable choice as a proxy for computationally expensive red-teaming.

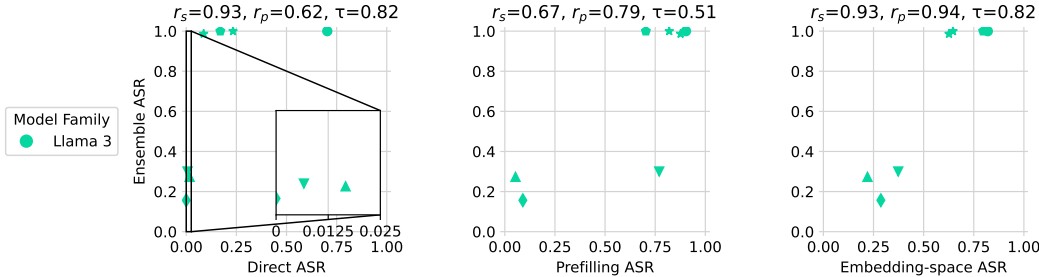

Figure 1: Attack success rates for different variants of Llama-3-8B. We include instruct versions (★) as a baseline and compare to safety-tuned (●), adversarially trained (▲,▼), circuit breaker (◆), and capability-optimized (●,●) models.

## 4.2 COMPARING ACROSS MODEL FAMILIES

We also investigate whether proxy methods can be used to predict the success rate of expensive red-teaming attacks on newly introduced model families. In Fig. 2, each point corresponds to a specific model from one of six model families (Gemma 2 (Team et al., 2024), Mistral (Jiang et al., 2023), Qwen (Bai et al., 2023), Phi-3 (Abdin et al., 2024), Llama 3 (Dubey et al., 2024), Llama 2 (Touvron et al., 2023)). Prefilling and embedding space attacks often have much higher ASR than direct prompting, which naïvely use the harmful prompt without any modification. Direct ASR is generally below $5\%$, except for models that are extremely unrobust (ensemble ASR close to $100\%$). Thus, the pairs of direct and ensemble ASR do not admit a linear fit and the Pearson correlation $r_p$ is small. However, the rank correlation coefficients of direct prompting ($r_s = 0.94$, $\tau = 0.83$) are higher than those of the other two proxy methods ($r_s = 0.79$, $\tau = 0.61$) and ($r_s = 0.90$, $\tau = 0.73$).

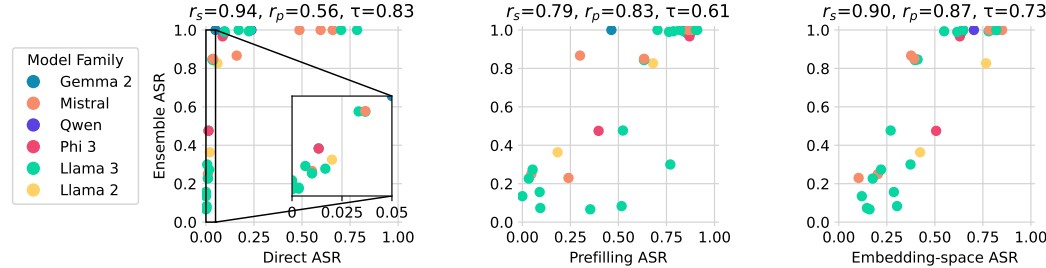

Figure 2: Attack success rates for models from different families. Direct ASR has the largest $r_s$ and $\tau$, i.e., the order of two models w.r.t. direct ASR is most predictive of order w.r.t. ensemble ASR.

## 4.3 ASSESSING EFFECTIVENESS OF ROBUSTNESS FINE-TUNING

A standard method for increasing model robustness is via post-training/fine-tuning approaches, e.g., via circuit breaker training (Zou et al., 2023) or continuous adversarial training (Sheshadri et al., 2024; Xhonneux et al., 2024). In Fig. 3 & 7, we assess whether proxy ASR can potentially be used to predict ensemble ASR after fine-tuning for a specific number of steps, rather than performing computationally expensive red-teaming for every possible value of this hyper-parameter. Specifically, we apply circuit breaker training to Llama-3-8B-Instruct and vary the number of training steps between 1 and 300. Again, while the relation between proxy ASR and ensemble ASR is generally monotonic and linear for all three proxies, direct prompting achieves significantly higher ranking correlations $r_s$ and $\tau$.

## 4.4 SCALING TRENDS

We find that the effectiveness of different proxy methods varies with the amount of prompts used (Fig. 4). Prefilling and embedding space attacks attain universally higher Pearson correlation, i.e., admit a better linear fit irrespective of the number of prompts. They can also reach higher Spearman and Kendall ranking correlation — but only when using few prompts. For 50 or more prompts, direct prompting yields higher ranking correlation coefficients. This can be explained as follows: Since

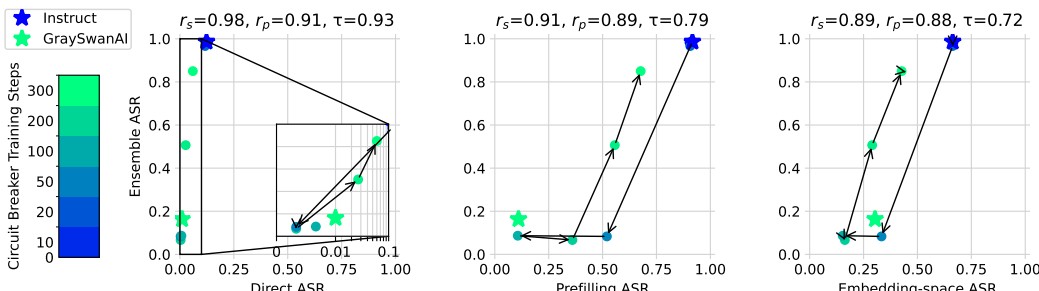

Figure 3: Attack success rates for different number of robustness fine-tuning steps using Circuit Breakers (Zou et al., 2024). We include the base instruct model and the officially released circuit breaker model. Despite varying success rate, all proxy methods have similar correlation coefficients, i.e., are similarly predictive of fine-tuning effectiveness. Arrows indicate training progression.

direct ASR is generally small for robust models, there is a high chance that our sample estimate will incorrectly indicate a direct ASR of exactly 0 when using few prompts, making the observed relation to ensemble ASR very erratic. Using more prompts provides a better estimate of the small but non-zero population success rate of direct prompting, thus eliminating this issue and making direct ASR a good predictor of whether one model will be more robust than another to our synthetic red-teamer. As increasingly robust models will decrease ASR, we expect to see an increase in the number of prompts required to effectively use direct prompting as a proxy.

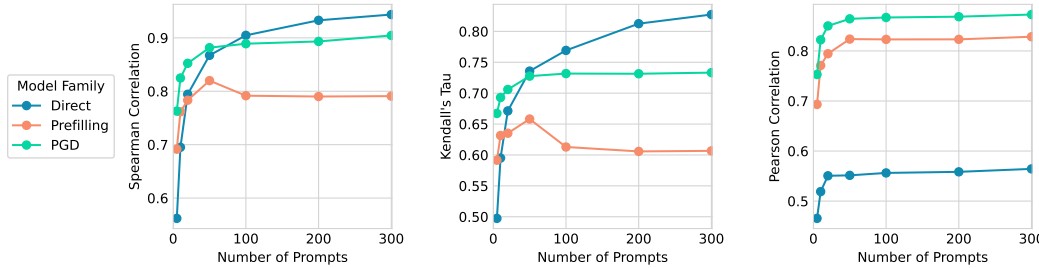

Figure 4: Correlation coefficients between proxy attack success rate and ensemble attack success rate under varying number of prompts. When using fewer than 50 prompts, PGD yields higher Spearman and Kendall ranking correlations, however the direct prompting scales better with more prompts. Prefilling and PGD achieve higher linear/Pearson correlations at any prompt count.

## 4.5 LIMITATIONS

While we conducted an exhaustive and computationally intensive evaluation using six attacks and 33 models from the sub-10B parameter class, our experiments should be further validated to ensure they generalize to other attack algorithms and model sizes. Moreover, concurrent work on efficient robustness evaluations through latent and weight manipulations observes less clear correlations between proxy and worst-case discrete attacks (Che et al., 2025).

## 5 CONCLUSION

We investigated the effectiveness of inexpensive proxy attacks in predicting LLM robustness against adversarial red-teaming. Our results highlight key trade-offs between different proxy methods. Direct prompting is a strong baseline for ranking models by robustness across diverse scenarios (within-family, cross-family, safety fine-tuning), provided that enough ($> 50$) prompts are used. Embedding-space attacks provide better ranking at low prompt count and better linear fits, while prefilling attacks are generally inferior to the two alternatives. Overall, our results showcase that efficient proxy attacks are a promising direction for future research towards making foundation models more responsible without incurring unjustifiable computational overhead.

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

## A  MODEL ZOO

| HuggingFace Model ID | Model Family | Category |
|---|---|---|
| google/gemma-2-2b-it | Gemma 2 | Instruct (★) |
| berkeley-nest/Starling-LM-7B-alpha | Mistral 7B | Capability-optimized (●) |
| cais/zephyr_7b_r2d2 | Mistral 7B | Safety Tuned (●) |
| ContinuousAT/Zephyr-CAT | Mistral 7B | Adv. Trained (▲) |
| GraySwanAI/Mistral-7B-Instruct-RR | Mistral 7B | Circuit Breaker (◆) |
| HuggingFaceH4/zephyr-7b-beta | Mistral 7B | Capability-optimized (●) |
| mistralai/Mistral-7B-Instruct-v0.3 | Mistral 7B | Instruct (★) |
| mistralai/Mistral-Nemo-Instruct-2407 | Mistral Nemo | Instruct (★) |
| mistralai/Ministral-8B-Instruct-2410 | Ministral | Instruct (★) |
| ContinuousAT/Llama-2-7B-CAT | Llama 2 | Adv. Trained (▲) |
| lmsys/vicuna-7b-v1.5 | Llama 2 | Capability-optimized (●) |
| meta-llama/Llama-2-7b-chat-hf | Llama 2 | Instruct (★) |
| LLM-LAT/robust-llama3-8b-instruct | Llama 3 | Adv. Trained (▼) |
| meta-llama/Meta-Llama-3-8B-Instruct | Llama 3 | Instruct (★) |
| NousResearch/Hermes-2-Pro-Llama-3-8B | Llama 3 | Capability-optimized (●) |
| GraySwanAI/Llama-3-8B-Instruct-RR | Llama 3 | Circuit Breaker (◆) |
| allenai/Llama-3.1-Tulu-3-8B-DPO | Llama 3.1 | Capability-optimized (●) |
| meta-llama/Meta-Llama-3.1-8B-Instruct | Llama 3.1 | Instruct (★) |
| meta-llama/Llama-3.2-1B-Instruct | Llama 3.2 | Instruct (★) |
| meta-llama/Llama-3.2-3B-Instruct | Llama 3.2 | Instruct (★) |
| qwen/Qwen2-7B-Instruct | Qwen2 7B | Instruct (★) |
| ContinuousAT/Phi-CAT | Phi 3 | Adv. Trained (▲) |
| microsoft/Phi-3-mini-4k-instruct | Phi 3 mini | Instruct (★) |
| microsoft/phi-4 | Phi 4 | Instruct (★) |

Table 2: List of models with their short names, base model family, and category. Sorted by family, then model ID.

In addition, we fine-tune Llama-3-8B-Instruct using the circuit breaker methodology (Zou et al., 2024) using $N = \{1, 10, 20, 50, 100, 200, 300, 500, 1000\}$ steps, and with the CAPO version of continuous adversarial training (Xhonneux et al., 2024) and $N = \{75, 150, 225\}$ steps. We use `bfloat16` quantization for all models.

## B  HYPERPARAMETERS & EXPERIMENTAL DETAILS

We run all attacks on all 300 harmful prompts from AdvBench (Zou et al., 2023), as included in HarmBench. A jailbreak attempt is counted as successful if both HarmBench's finetuned Llama-2-13B classifier (Mazeika et al., 2024) and LlamaGuard 3 8B (Dubey et al., 2024) flag the model's response as harmful.

The hyperparameters for the attacks used in the ensemble and the proxy attacks are shown below. Where possible, attack implementations were sourced from the original authors' GitHub repositories; otherwise, we integrated a HarmBench implementation into our pipeline. In some cases, we consulted authors directly to obtain reference implementations and verify correctness. For all attacks, we evaluate a single greedy generation per prompt-candidate.

- AmpleGCG Liao & Sun (2024): We use `osunlp/AmpleGCG-llama2-sourced-llama2-7b-chat` to generate 200 attack suffixes with diversity penalty 1 and generate completions for all 200 of the attack candidates.
- AutoDAN Liu et al. (2023): We use 100 steps and initialize using the 128 seed prompts from HarmBench's implementation. We use the attacked model itself as mutator model and set $N_{\text{elites}} = 0.05$, crossover $= 0.5$, $N_{\text{points}} = 5$, and $P_{\text{mutation}} = 0.01$.
- BEAST Sadasivan et al. (2024): We use $k1 = k2 = 15$ and set the temperature to 1 to sample $N = 40$ suffix tokens.
- GCG Zou et al. (2023): We use a modified version of nanoGCG with a corrected token filtering algorithm to remove special tokens & only allow ASCII-representable characters. We set $N_{\text{steps}} = 250$, use a batch size of 512, Top-K $= 256$, and initialize using the string "x x x x x x x x x x x x x x x x x x x x" as it tokenizes to exactly 20 tokens for all tested models.
- HumanJailbreaks: We use the 114 human-designed jailbreak templates in HarmBench Mazeika et al. (2024) to prompt the model.
- PAIR Chao et al. (2023): We use `lmsys/vicuna-13b-v1.5` as attacker model and generate up to 512 tokens per attack prompt. Sampling attacks is done with with temperature 1 and top-p of 0.9, setting $N_{\text{streams}}$ to 5 and $N_{\text{iterations}}$ to 5. During the attack, the victim model generates up to 256 tokens using greedy generation. If the conversations grow longer than the model's context, we truncate the first non-system messages from the conversation until the conversation fits into the context window.

The proxy attacks use the following settings:

- Direct: We simply use the harmful prompt without any modification and sample a greedy generation.
- Embedding-space Schwinn et al. (2024): We initialize the attack using the suffix "x x x x x x x x x x x x x x x x x x x x" as it tokenizes to exactly 20 tokens for all tested models, and run signed gradient descent optimization for 100 steps. We use a learning rate of $\alpha = 0.01$ and constrain the optimization to an $L_2$ ball with radius 1 around the initialization for each token. To normalize across model families, we normalize both step size and $L_2$ constaint by the average $L_2$ embedding norm across the input vocabulary.
- Prefilling: We use the unmodified harmful prompt and pre-fill the beginning of the model's response using the affirmative target sequence from the dataset.

Running the attack ensemble on an Nvidia H100 GPU for a single prompt requires 1,731 seconds on average, while direct prompting and prefilling can be easily batched and is completed in a single second. Batched embedding space attacks require approximately 5 seconds per prompt.

# C  ADDITIONAL EXPERIMENTAL RESULTS

## C.1  MISTRAL VARIANTS

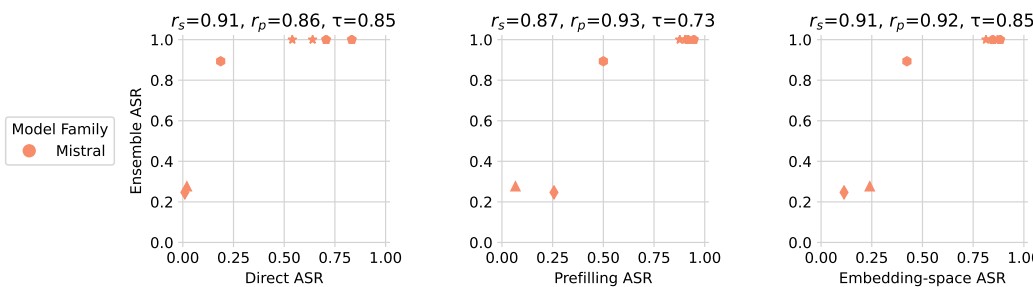

Figure 5: Attack success rates for different variants of Mistral 7B Instruct. We include instruct versions (★) as a baseline and compare to safety-tuned (●), adversarially trained (▲), circuit breaker (♦), and capability-optimized (⬡) models.

## C.2  LLAMA 3 VARIANTS

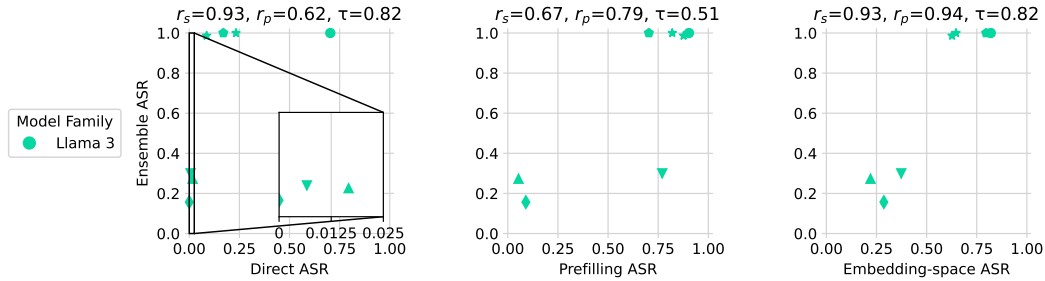

Figure 6: Attack success rates for different variants of Mistral 7B Instruct. We include instruct versions (★) as a baseline and compare to safety-tuned (●), adversarially trained (▲), circuit breaker (♦), and capability-optimized (⬡) models.

These model families were selected due to their popularity and resulting large number of versions.

## C.3  CONTINUOUS ADVERSARIAL TRAINING

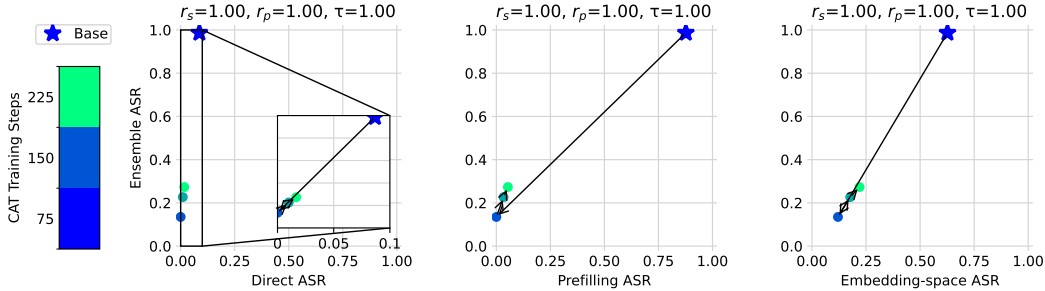

Figure 7: Attack success rates for different number of robustness fine-tuning steps using Continuous Adversarial Training (Xhonneux et al., 2024) on Llama 3 8B Instruct. All methods are highly correlated with the synthetic red-teamer. Due to resource and time constraints we only compare four training checkpoints. Arrows indicate training progress.

