# OpenReview forum: "Fast Proxies for LLM Robustness Evaluation"
_ICLR.cc/2025/Workshop/BuildingTrust — BuildingTrust_

### Official Review · Reviewer_TM7b · 2025-03-01
**This paper explores fast proxy methods for LLM robustness but lacks broader validation across models and attack techniques**

**Rating:** 5
**Confidence:** 2

**Review:**

## Summary
This paper explores computationally efficient proxy methods for assessing the robustness of large language models (LLMs) against adversarial attacks. The key claim is that these proxies—direct prompting, prefilling, and embedding-space attacks—can serve as reliable indicators of a model’s susceptibility to adversarial red-teaming without requiring costly attack methods. The authors construct a strong synthetic red-teamer by combining six adversarial attack algorithms and use this as a benchmark to evaluate the proxy methods across 33 open-source models.

Their findings suggest that while direct prompting does not achieve high absolute attack success rates (ASR), it correlates well with the more expensive attack ensemble when a sufficient number of prompts (>50) are used. Embedding-space attacks demonstrate a stronger correlation with ensemble methods in low-prompt settings but require white-box access, making them impractical for many real-world applications. Prefilling attacks, which prepend adversarial content to a model’s output, generally perform worse than the other two approaches.

## Strengths
- Broad Model Selection – The study covers a diverse set of models spanning multiple families (Llama, Mistral, Qwen, Phi, etc.), offering valuable insights into robustness trends both within and across model families.
- Computational Efficiency – The proposed proxy methods reduce evaluation costs by several orders of magnitude compared to traditional red-teaming, making them attractive for large-scale safety assessments.
- Scalability Analysis – The authors examine how the effectiveness of different proxy methods scales with the number of prompts, providing practical guidance for researchers balancing accuracy with efficiency.
- Trade-Offs Between Methods – The comparative analysis highlights the strengths and limitations of each proxy approach, particularly in terms of feasibility in real-world attack scenarios.

## Weaknesses
- Limited Exploration of Proxy Methods – The study evaluates only three proxy techniques, while other techniques (e.g., adaptive adversarial prompting [1], reinforcement learning-based attackers [2]) could provide additional insights.
- Justification for Direct Prompting’s Suitability – While the authors conclude that direct prompting is a strong baseline, their own data show that embedding-space attacks correlate better with the expensive red-teaming ensemble (particularly within model families). The preference for direct prompting is argued on the basis of real-world feasibility, but this could be better substantiated.
- Focus on Small-Scale Models – The study is limited to models under 10B parameters, leaving open the question of whether these proxy methods would generalize to frontier-scale LLMs, where adversarial robustness is a more pressing issue.
- Terminology and Clarity – The paper introduces several model categorization terms (e.g., "instruct," "safety-tuned," "adversarially trained," "circuit breaker," and "capability-optimized") without clearly defining or referencing them. Additionally, technical terms like PGD and ASR are used without definition, making it harder for readers unfamiliar with adversarial attack literature to follow the work. Also, while the correlation coefficients are well presented, some visualizations do not effectively communicate key insights, particularly regarding the relative strengths of different proxy methods.

## Overall Evaluation
The paper addresses an important problem in adversarial robustness evaluation by proposing faster, cheaper alternatives to traditional red-teaming. Its broad empirical validation is a strength, but the analysis could be more comprehensive, particularly in justifying the ranking of proxy methods and expanding to larger models. Future work should expand to larger models and include additional proxy techniques to provide a more comprehensive picture of lightweight adversarial robustness testing.

## References
[1] Paulus, A., Zharmagambetov, A., Guo, C., Amos, B., & Tian, Y. (2024). Advprompter: Fast adaptive adversarial prompting for llms. arXiv preprint arXiv:2404.16873.
[2] Wang, X., Peng, J., Xu, K., Yao, H., & Chen, T. (2024, August). Reinforcement learning-driven llm agent for automated attacks on llms. In Proceedings of the Fifth Workshop on Privacy in Natural Language Processing (pp. 170-177).

---

### Official Review · Reviewer_kNqG · 2025-03-01
**Review: Fast Proxies for LLM Robustness Evaluation**

**Rating:** 7
**Confidence:** 4

**Review:**

The paper "Fast Proxies for LLM Robustness Evaluation" proposes a computationally efficient framework for evaluating the robustness of LLMs to adversarial attack. The authors use relatively inexpensive proxies to estimate LLM robustness, which include direct prompting, embedding-space attacks, and pre-filling. The authors demonstrate that these proxy methods can predict ASRs with high accuracy, offering a promising approach for scaling robustness evaluations. The experiments cover a wide range of LLMs, providing a comprehensive set of results. Slightly concerning is the lack of analysis into false negatives/positives which could skew results. While the correlation results between proxies and full attack success rates are compelling, the explanation of this phenomena is not well covered. More insight into the underlying mechanisms of the proxies would help understanding adversarial robustness in LLMs. The paper is clearly written, and the work is of significant practical importance - the study's findings could significantly reduce the cost of adversarial testing in LLMs, making it more feasible to conduct safety evaluations.

---

### Official Review · Reviewer_HzLC · 2025-03-02
**Effective Proxy Metrics for Efficient LLM Robustness Evaluation**

**Rating:** 7
**Confidence:** 3

**Review:**

review


The paper presents a systematic investigation into the efficacy of low-cost proxy methods—direct prompting, prefilling attacks, and embedding-space attacks—for predicting the robustness of large language models (LLMs) against computationally intensive adversarial red-teaming. By leveraging a synthetic red-teamer ensemble comprising six distinct attack algorithms and evaluating over 7 million jailbreak attempts across 33 open-source models, the authors demonstrate that proxy metrics exhibit strong correlations with full attack ensemble outcomes.

Quality

The experimental framework is rigorous, encompassing diverse model families (Llama, Mistral, Phi, etc.) and evaluation scenarios (within-family, cross-family, and robustness fine-tuning). The use of standardized metrics (Pearson/Spearman correlations) and large-scale datasets (300 harmful prompts from AdvBench) ensures statistical validity. The inclusion of multiple checkpoints during adversarial training (e.g., Circuit Breaker and Continuous Adversarial Training) adds depth to the analysis. However, the study’s exclusive focus on sub-10B parameter models limits insights into the generalizability of proxy methods to larger architectures (e.g., Llama-3-70B).

Clarity

The manuscript is well-organized, with a logical progression from problem definition (high costs of red-teaming) to methodology and results. Figures such as Figure 2 (cross-family model correlations) and Figure 4 (scaling trends with prompt count) effectively visualize key findings. The appendix provides additional technical details, though some sections (e.g., attack algorithm descriptions) could benefit from conciseness.

Originality

This work addresses a critical gap in LLM safety evaluation by proposing cost-effective alternatives to resource-intensive red-teaming. Key innovations include the first systematic validation of proxy methods, revealing that direct prompting—despite its simplicity—serves as a strong baseline for ranking model robustness. Additionally, the study demonstrates the utility of proxy metrics for cross-family robustness prediction and dynamic monitoring of robustness improvements during adversarial fine-tuning.

Significance

The findings have substantial practical implications for LLM safety research. By reducing evaluation costs by ~1,000x, proxy methods democratize robustness assessments for resource-constrained teams. The correlation between proxy metrics and adversarial training trajectories (e.g., Circuit Breaker training) offers actionable insights for model alignment strategies.

Strengths

The study’s strengths lie in its scale and methodological rigor, spanning 33 models, 6 attack algorithms, and 7 million jailbreak attempts. Direct prompting emerges as a particularly practical baseline due to its simplicity and accessibility. The multi-scenario analysis—encompassing within-family comparisons, cross-family generalization, and training-phase monitoring—underscores the versatility of proxy methods.

Weaknesses

Limitations include the focus on sub-10B parameter models, which raises questions about the generalizability of proxies to larger architectures. The synthetic attack ensemble, while comprehensive, lacks diversity in newer attack paradigms (e.g., multi-turn or multimodal jailbreaking). Direct prompting’s sensitivity to prompt count (>50 required for stable correlations) may hinder its applicability in low-resource settings. Furthermore, embedding-space attacks rely on white-box assumptions, diverging from real-world black-box threat models.

---

### Decision · Program_Chairs · 2025-03-04

Accept